# Radiodynamic Therapy for High-Grade Glioma in Normoxic and Hypoxic Environments for High-Grade Glioma

**DOI:** 10.3390/cancers17243927

**Published:** 2025-12-08

**Authors:** Erika Yamada, Eiichi Ishikawa, Tsubasa Miyazaki, Hirofumi Matsui, Kazuki Akutagawa, Masahide Matsuda, Alexander Zaboronok, Hiroshi Ishikawa

**Affiliations:** 1Department of Neurosurgery, Institute of Medicine, University of Tsukuba, Tsukuba 305-8576, Japan; erika26561@gmail.com (E.Y.); akutagawa.kazuki.kq@ms.hosp.tsukuba.ac.jp (K.A.); m-matsuda@md.tsukuba.ac.jp (M.M.); a.zaboronok@md.tsukuba.ac.jp (A.Z.); 2Laboratory of Clinical Regenerative Medicine, Department of Neurosurgery, Institute of Medicine, University of Tsukuba, Tsukuba 305-8576, Japan; ishi-hiro.crm@md.tsukuba.ac.jp; 3Cell-Medicine, Inc., Sengen 2-1-6, Tsukuba Science City, Tsukuba 305-0047, Japan; t-miyazaki@cell-medicine.com; 4Department of Gastroenterology, Institute of Medicine, University of Tsukuba, 1-1-1 Tennodai, Tsukuba 305-8575, Japan; hmatsui@gwe.md.tsukuba.ac.jp

**Keywords:** radiodynamic therapy, 5-ALA, glioma, hypoxia, immune reaction

## Abstract

This study investigated the therapeutic potential of radiodynamic therapy (RDT), which combines the administration of the photosensitizer 5-aminolevulinic acid (5-ALA) with X-ray irradiation, for high-grade glioma (HGG). In vitro experiments using U87MG and U251MG glioma cell lines demonstrated that RDT was effective under normoxia (20% oxygen), significantly reducing cell viability in a concentration-dependent manner and increasing reactive oxygen species (ROS) production. However, RDT efficacy was diminished in hypoxic environments. RNA-seq analysis revealed that RDT in normoxia affected immune reaction-related genes, while RDT in hypoxia modulated genes related to epithelial–mesenchymal transition (EMT) and suppressed increases in the angiogenesis marker vascular endothelial growth factor (VEGF). These findings suggest that RDT holds promise as a therapeutic strategy for HGG, particularly in normoxic conditions by increasing ROS production and enhancing immune responses. It may also be advantageous under hypoxia, at least in terms of inhibiting angiogenesis.

## 1. Introduction

Radiodynamic therapy (RDT) is a novel, minimally invasive therapeutic approach that involves the selective intracellular accumulation of a photosensitizer within tumor tissue, followed by X-ray irradiation of the targeted tumor area [1]. Photodynamic therapy (PDT), a clinically established treatment, similarly utilizes preoperative administration of a photosensitizer, such as talaporfin sodium (TS, or mono-L-aspartyl chlorin e6, NPe6), with high tumor specificity, followed by intraoperative laser irradiation of the tumor cavity.

One advantage of using X-rays instead of visible light to excite photosensitizers is that X-rays can be easily applied externally, even after surgery, whereas lasers can only be used intraoperatively during a craniotomy to irradiate the cavity, potentially targeting remnant tumor cells that have not yet formed a visible tumor on the surface. Protoporphyrin IX (PpIX), an intracellularly produced photosensitizer when patients receive 5-aminolevulinic acid (5-ALA), is also known to respond to irradiation [2]. Studies indicate that X-ray irradiation of PpIX generates reactive oxygen species (ROS) within mitochondria, with damaged mitochondria initiating chain reactions that induce cell death through sustained oxidative stress [3,4,5]. Additionally, 5-ALA-RDT induces cell cycle arrest, increases oxidative stress, and activates anti-tumor immunity [6]. Specifically, 5-ALA enhances TNF-α production in glioma cell lines and suppresses prostaglandin E2 (PGE2) production via cyclooxygenase-2 (COX-2) and microsomal PGE2 synthase-1 (mPGES-1), thereby counteracting immune evasion mechanisms in macrophages and promoting an anti-tumor effect [7].

Another advantage is that radiation therapy is an established postoperative treatment modality for high-grade glioma (HGG). Currently, the standard treatment for HHGs includes chemo-radiotherapy (CRT), which involves 60 Gy irradiation in conjunction with the chemotherapeutic agent temozolomide (TMZ) following maximal surgical resection [8]. Oral administration of 5-ALA alongside irradiation for the purpose of radiodynamic effect induction, named 5-ALA-RDT, is expected to enhance the anti-tumor effect [9,10,11]. Considering this radiodynamic induction, a Phase I/II trial of 5-ALA-RDT is underway [12], although results are yet to be reported.

Additionally, 5-ALA can theoretically be administered multiple times before each radiotherapy session, presenting a practical and repeatable treatment option. However, a limitation of RDT compared to PDT is that it does not occur immediately after tumor removal under normoxic conditions. It is known from pathological findings that the environment in which brain tumors are located is always hypoxic, as the growth of blood vessels cannot keep up with the growth of the tumor [13]. The hypoxic conditions cause brain tumors to acquire active migratory ability, invasiveness, and resistance to chemotherapy and radiotherapy [13,14,15]. Currently, no studies compare changes in gene and protein expression in cells after RDT under normoxic and hypoxic conditions. Thus, we conducted experiments to assess 5-ALA-RDT in both normoxic and hypoxic environments, aiming to verify its therapeutic effects when 5-ALA is administered during radiation therapy for HGG. We also aimed to explore its mechanism, impact on the immune microenvironment, and potential in overcoming radioresistance.

## 2. Materials and Methods

Glioma cell lines U87MG and U251MG were obtained from the American Type Culture Collection (Rockville, MD, USA). Glioma cell lines T98G were purchased from the RIKEN cell bank (Tsukuba, Japan). Hypoxia-adapted U87MG cells (U87MG_long_hypo_ cells) were generated by culturing U87MG cells in an incubator with 3% O_2_ for approximately 6 months. All cell lines were cultured in DMEM and passaged every 3–4 days. U87MG cells at passage number ≤20 (*p* ≤ 20) were used in the experiments. 5-ALA was administered in the form of the pharmaceutical product Alabel^®^. X-ray irradiation was performed using a Hitachi Power Solutions MBR-1520R-3 system at 120 kV and 5 mA, with an aluminum (Al, 0.5 mm) and copper (Cu, 0.1 mm) shielding plate.

### 2.1. Enhancement Effect of RDT by 5-ALA Concentration and X-Ray Dose

To evaluate the effect of RDT in a 20% O_2_ environment, U87MG cells maintained in 20% O_2_ (U87MG_normox_ cells) were seeded in a 12-well plate, and the culture medium was replaced with DMEM containing 0, 250, 500, or 1000 μM 5-ALA. The cells were cultured in an incubator at 37 °C and 5% CO_2_ in the dark. X-ray irradiation was applied at various doses 4 h after 5-ALA administration in 20% O_2_ environment. For RDT under hypoxic conditions, U87MG cells were cultured in an incubator with oxygen concentrations set to 3% and 1% (U87MG_3%hypox_ and U87MG_1%hypox_ cells) to represent chronic hypoxia within the brain environment. After seeding and adding 5-ALA, the cells were incubated at the desired oxygen concentration before X-ray irradiation. U87MG_long_hypo_ cells were also used to examine RDT efficacy under conditions similar to those in the previous experiments with U87MG cells.

Regarding 5-ALA and irradiation doses, the efficacy of RDT was assessed using a cell counting method. U87MG cells were treated with 5-ALA at concentrations of 0 to 1000 μM, followed by X-ray irradiation. In the initial experiment (Figure 1), the cells were irradiated with X-rays at doses 2–20 Gy, and in subsequent experiments, a regimen of 2 Gy X-rays once a day for 3 days (6 Gy in total) was used. Cell counts were taken 7 days after irradiation, either using a cell counter or by measuring absorbance at 450 nm using Cell Counting Kit-8 (FUJIFILM Wako Pure Chemical Corporation, Osaka, Japan) and a plate reader. To assess RDT efficacy using a colony formation assay, U87MG cells were treated with or without 5-ALA and then irradiated with 2 Gy X-rays once a day for 3 days (6 Gy in total). Fourteen days after irradiation, colonies were counted to evaluate the ability of RDT to inhibit colony formation.

For intracranial experiments, U87 cells were transplanted intracranially into BALB/c-nu/nu mice (CLEA Japan, Inc., Tokyo, Japan) to form tumors. The skin was incised, and a burr hole was drilled 1 mm anterior and 2 mm lateral to the fontanelle. The needle tip was fixed at a depth of 3 mm below the brain surface, and tumor cells were slowly injected using a glass syringe. Following tumor engraftment, 5-ALA (20 mg/kg) was administered orally. Two hours after administration, the tumor was irradiated with radiation (2 Gy). This procedure was repeated three times, delivering a total dose of 6 Gy. Tumors were excised 1–2 weeks post-irradiation and subjected to immunohistochemical staining for PD-L1 (28-8, Abcam) and VEGF (VG-1, Abcam). The expression of PD-L1 or VEGF in tumor cells was assessed using staining scores of “-” (no staining), “+” (less than 25% stained cells), “++” (25–50% stained cells), or “+++” (>50% stained cells) [16].

### 2.2. Changes in the Intracellular 5-ALA (PpIX) Under Hypoxic Conditions

The intracellular levels of PpIX induced by 5-ALA administration with irradiation in 20% O_2_ and 3% O_2_ environments were analyzed by measuring fluorescence intensity. U87MG cells were seeded in 12-well plates, and the culture medium was replaced with DMEM containing 0 or 1000 μM of 5-ALA. The cells were then incubated in either 20% O_2_ or 3% O_2_ at 37 °C with 5% CO_2_ in the dark. Four hours after adding 5-ALA, the cells were irradiated with 2 Gy X-rays once a day for three days (6 Gy in total). After completing irradiation, the medium was removed, and the cells were washed with PBS and lysed using RIPA buffer. The lysate was transferred to a 96-well cell culture plate, and the fluorescence was measured using a microplate reader (excitation at 415 nm, emission at 625 nm).

### 2.3. Changes in Intracellular ROS Due to 5-ALA Administration

The culture medium from U87MG_normox_, U87MG_3%hypox_, or U87MG_long_hypo_ cells was replaced with the medium containing 0 or 1000 μM of 5-ALA. Si-DMA for Mitochondrial Singlet Oxygen Imaging (FUJIFILM Wako) was used according to the manufacturer’s instructions. To prepare a 100 μmol/L Si-DMA DMSO stock solution, 2 μg of Si-DMA was added to 36 μL of DMSO. Then, 9.6 μL of this stock solution was mixed with 24 mL of HANKS (HBSS) to create a 40 nmol/L Si-DMA working solution. Four hours after the 5-ALA addition, the supernatant was removed, and the cells were washed twice with HANKS. 3 mL of 40 nmol/L Si-DMA working solution was added to each dish, and the cells were incubated for 45 min. The supernatant was then removed, and the dish was washed twice with HANKS. PBS was added, and singlet oxygen in the mitochondria was observed under a fluorescence microscope (excitation at 600 ± 25 nm, emission at 685 ± 25 nm).

### 2.4. Mechanism of RDT Effect on Tumor Cells and the Micro-Immune Environment

For in vitro experiments, U87MG cells were cultured for several days in a 20% O_2_ environment or long-term in a 3% O_2_ environment. RDT was conducted under 20% O_2_ or 3% O_2_ conditions using the same protocol described above, with or without 1000 μM of 5-ALA and/or 6 Gy X-ray irradiation (delivered as 2 Gy per session for three sessions). After treatment, RNA was extracted from cells for sequencing, and supernatants were collected for analysis using the ELISA method.

### 2.5. Enzyme-Linked Immunosorbent Assay (ELISA)

The levels of various proteins in the supernatant of U87MG cell culture under different conditions were measured using ELISA kits according to the manufacturers’ protocols. For HIF-1α, the TMB ELISA Substrate (ab171527, Abcam, Cambridge, UK) was used, while VEGF (ab222510, Abcam, Cambridge, UK), soluble PD-L1 (ab277712, Abcam, Cambridge, UK), IL-10 (ab185986, Abcam, Cambridge, UK), LIF (ab242228, Abcam, Cambridge, UK), Nrf2 (ab277397, Abcam, Cambridge, UK), and CHO HCP (ab240996, Abcam, Cambridge, UK) concentrations were also measured using kits from Abcam. Additional kits included the CHO HCP ELISA kit (ab240996, Abcam, Cambridge, UK) for HCP-1, the Human CSF-1 ELISA kit (Cat. Nos. EHCSF1 and EMCSF1; Thermo Fisher Scientific, Waltham, MA, USA) for CSF-1, the Human HMGB1 ELISA kit (Cat. No. KE00345, Proteintech, Rosemont, IL, USA) for HMGB1, and the Human CRT ELISA kit (Cat. No. E-EL-H0627, Elabscience, Wuhan, China) for CRT.

### 2.6. RNA Sequencing (RNA-Seq) on Tumor Cells

RNA-seq was conducted on U87MG_normox_ and U87MG_3%hypox_ cells. For each condition, 5 × 10^5^ cells were collected, lysed with QIAzol Lysis Reagent, and stored at −80 °C. The sequencing was performed using the Illumina NextSeq500 platform (Illumina, Inc., San Diego, CA, USA) at the Sports Medicine Analysis Division, Office of the Open Facilities Network, the University of Tsukuba, according to the standard protocol [17], with 36-base paired-end reads. In this RNAseq experiment, one dataset was obtained for each sample to determine the trend with each treatment, and the median values of 4 data sets for the normoxia group and 4 data sets for the hypoxia group were compared.

After sequencing, FASTQ files were exported, and basic quality metrics for the NGS run data were assessed using CLC Genomics Workbench 24.0 (QIAGEN GmbH, Hilden, Germany). Quality assessment confirmed that 99.77% of all reads had a PHRED score of 20 or higher, indicating a successful run. The number of paired-end reads per sample ranged from approximately 25.91 million to 50.33 million. Gene expression analysis was performed using RNAseqChef (https://kan-e.github.io/RNAseqChef/, accessed on 1 May 2025).

## 3. Results

### 3.1. Enhancement Effect of RDT by 5-ALA Concentration and X-Ray Dose

In a 20% O_2_ environment, the number of U87MG_normox_ cells decreased significantly in a concentration-dependent manner with 5-ALA at 2 Gy and 6 Gy; however, the RDT effect remained nearly constant regardless of 5-ALA concentration at doses above 8 Gy (Figure 1A). The combination of 5-ALA and X-rays (RDT) was found to be effective in the range of at least 2 Gy to 10 Gy in a 20% O_2_ environment. On the other hand, in a 3% O_2_ or long-term 3% O_2_ environment, the RDT effect, as measured by cell count, was not dependent on 5-ALA concentration and showed no significant difference compared to the control (no 5-ALA) group across various 5-ALA concentrations at each irradiation dose. This phenomenon was reproduced in a second experiment (Figure 1B–E).

To evaluate changes in intracellular 5-ALA (PpIX) under hypoxic conditions, the fluorescence intensity of PpIX was higher when 5-ALA was administered in a 3% O_2_ environment compared to a 20% O_2_ environment. Both the 3% O_2_ 5-ALA and RDT groups showed greater fluorescence intensities than the 20% O_2_ groups (Figure 1F), indicating that the hypoxic environment did not reduce PpIX accumulation. Therefore, we subsequently observed intracellular ROS using fluorescence imaging.

The difference in the RDT effect at the same set of oxygen environments was further verified using a colony formation assay (Figure 1G,H). In a 20% O_2_ environment, colony numbers tended to decrease with the combination of 5-ALA administration and X-ray irradiation compared to 5-ALA alone or X-ray irradiation alone. In both short-term and long-term 3% O_2_ conditions, however, colony numbers remained the same for X-ray irradiation alone and for the combination with 5-ALA. In U87MG_long_hypo_ cells, the inhibitory effect of radiation on colony formation tended to be weaker than in other cells. Similar results were also observed in U251MG cells and T98G cells (Figure 1I).

### 3.2. Changes in Intracellular ROS Due to 5-ALA Administration

To determine whether intracellular ROS contribute to the anti-tumor effect of 5-ALA, U87MG cells were treated with 5-ALA in both 20% O_2_ and 3% O_2_ environments, and ROS levels were evaluated 4 h after 5-ALA administration using fluorescence microscopy (Figure 2). In this and subsequent experiments the concentration of 1000 μM was adopted with reference to past papers on other carcinomas [11,18] and from the results of experiment in Figure 1. In the U87MG_normox_ cells, 5-ALA administration resulted in an increase in singlet oxygen compared to the control without 5-ALA. On the other hand, in U87MG_3%hypox_ and U87MG_long_hypo_ cells, the increase in ROS was minimal even with 5-ALA administration.

### 3.3. Comprehensive Search for Factors Affecting RDT in RNA-Seq

Using U87MG cells, we conducted in vitro experiments across four groups, with or without 5-ALA and/or X-rays, and comprehensively searched for genes expressed under each condition using RNA-seq (Appendix A). MA-plot, Volcano plot, and enrichment analysis comparing untreated (5ALA−/RT−) U87MG_normox_ and U87MG_3%hypox_ cells showed that hypoxia upregulated genes associated with epithelial–mesenchymal transition (EMT) and hypoxia-related gene expression compared to normoxia (Figure 3A,B). Under 20% O_2_, RDT induced changes in immune reaction-related and various other genes, while under 3% O_2_, RDT influenced EMT-related genes but not immune reaction-related genes (Figure 3C–F). Among these EMT-related genes, DNA-damage-inducible transcript 4 (DDIT4), a negative regulator of the mammalian target of rapamycin (mTOR), was notably upregulated under 3% O_2_ and RDT conditions. DDIT4 was highlighted in enrichment analysis for both the hypoxic control (Figure 3B) and hypoxic RDT (Figure 3D) conditions, suggesting it may be a key molecule in this context.

A color heat map of representative ischemia- and angiogenesis-related genes showed that most of these genes, including HIF1A, were generally upregulated under hypoxia. However, RDT under both normoxic and hypoxic conditions was shown to reduce VEGF expression compared to RT alone (Figure 3G). In contrast, representative inflammatory response genes were activated by RDT under normoxia but were less responsive under hypoxia (Figure 3H). In the EMT-related gene set, most EMT-related genes are upregulated in hypoxia, with some clusters further upregulated or downregulated by RDT (Figure 3I).

To validate these findings, we measured levels of representative ischemia- and angiogenesis-related proteins, as well as representative inflammatory response proteins using ELISA.

### 3.4. Mechanism of RDT Effect on Tumor Cells and the Micro-Immune Environment

In vitro experiments were conducted on U87MG and U251MG cells, which were divided into four groups: with or without 5-ALA and X-rays. Concentration of each substance was measured using the ELISA method (Figure 4). In a 20% O_2_ environment, the secretion of soluble PD-L1 (sPD-L1) was reduced by the combination of both 5-ALA and X-rays. The angiogenesis markers, soluble HIF-1α and VEGF, increased in the 3% O_2_ environment and with 5-ALA monotherapy, but the increases were suppressed in the combined treatment group.

Additionally, the secretion of colony-stimulating factor-1 (CSF-1), Leukemia Inhibitory Factor (LIF, a differentiation inhibitory marker for macrophages), and IL-10 (a cytokine that induces a dominant immunosuppressive macrophage phenotype) increased in the 20% O_2_ environment with 5-ALA administration. The IL-10 secretion tended to decrease in the combined treatment group under 20% O_2_ in both U87MG and U251MG cells. The secretion of CSF-1 tended to be similarly reduced in the combined treatment group under 20% O_2_ only in U251MG cells.

For Nrf2, an anti-inflammatory marker, expression was suppressed under 20% O_2_ by 5-ALA alone, radiation alone, or their combination. However, under hypoxic conditions (3% O_2_ or 1% O_2_), Nrf2 increased with 5-ALA administration and was suppressed by radiation in the 3% O_2_ environment. The expression of calreticulin (CRT) and high mobility group box-1 (HMGB1), which are tumor immunity activators, increased with 5-ALA alone in 3% O_2_ environment but was suppressed by X-ray irradiation across all environments.

### 3.5. VEGF and PD-L1 Alterations in Intracranial Model

For in vivo experiments using U87MG cells (Figure 5), U87MG cells were intracranially transplanted, and the mice were divided into two groups: a control group and an RDT group treated with 5-ALA and X-ray. The tumors were excised and immunostained. In the control group, both VEGF and PD-L1 showed strong positivity, whereas in the RDT group, VEGF expression was suppressed.

## 4. Discussion

Cell death caused by X-ray irradiation is generally thought to be due to both the direct effect of DNA damage and the indirect effect of ROS produced [11,19]. In a 20% O_2_ environment of our study using glioma cell line, intracellular ROS increased due to 5-ALA administration, suggesting that X-ray irradiation would further elevate ROS levels, thereby promoting a synergistic effect on cell death. However, in a 3% O_2_ environment, the low oxygen levels limit ROS generation even with 5-ALA administration, and the cell damage caused by X-ray irradiation and ROS is minimal. These results differed from those of previous studies with prostate cancer cells that showed 5-ALA overcomes hypoxia-induced radiation resistance by enhancing mitochondrial reactive oxygen species production [11]. For this reason, we speculate as follows; at first, the duration of 5-ALA exposure is different between this previous study and ours, although the concentration of 5-ALA is the same. Also, the cell types are very different: prostate cancer cells in the previous paper and glioma cells in our paper. This dissociation of results may be a biological difference between prostate cancer, which is relatively normally exposed to oxygen, and gliomas, which grow under hypoxia.

From the perspective of tumor cell death or growth inhibition through RDT in vitro, we believed that experimental conditions would have little effect on reducing cell numbers or inhibiting colony formation. However, as both radiation therapy and RDT induce changes in the expression of various molecules in both 20% O_2_ and 3% O_2_ environments, they likely influence the expression of various genes and proteins that affect the tumor microenvironment in an oxygen-dependent manner. This point will be discussed in further detail below.

### 4.1. Expression of PD-L1 with 5-ALA Administration

In this study, we observed an increase of PDL1 gene expression under hypoxia and a suppression of PD-L1 gene expression with RDT. Previous studies have shown that 5-ALA administration induces ROS production, which in turn can increase PD-L1 expression [20,21]. Furthermore, it has also been reported that PD-L1 expression can be upregulated by X-ray irradiation [22]. Thus, the increase in PD-L1 expression observed with RDT in this study can be theoretically explained, although it should be noted that the concentration of 5-ALA used in this study (1000 μM) is higher than that currently used in clinical practice.

High PD-L1 expression is generally associated with treatment resistance, as it can reduce the antitumor activity of cytotoxic T cells in the tumor microenvironment. Therefore, based on the results, it could also be considered that combining RDT and PD-L1 antibody therapy after irradiation could enhance antitumor effects. Furthermore, our ELISA experiments on immune microenvironment-related proteins after RDT revealed that irradiation suppressed the increased secretion of sPD-L1 caused by 5-ALA administration, especially in a 20% O_2_ environment. sPD-L1 is known to contribute to cytotoxic T cell exhaustion [23]. Although there are no specific studies on glioblastoma (GBM), it is known that high levels of sPD-L1 in the blood are associated with poor prognosis in non-small cell lung cancer and diffuse large B-cell lymphoma [24,25]. These results indicate that RDT may suppress tumor immune escape mechanisms mediated by sPD-L1 secretion.

### 4.2. RDT Potential Under Normoxia

The results of the present study suggest that RDT in a normoxic environment may help overcome treatment resistance (Figure 6), primarily by influencing immune response-related molecules. CSF-1, produced by tumor cells, is known to induce monocytes in the peripheral blood to differentiate into tumor-associated macrophages (TAMs) [26]. LIF, a cytokine originally identified as an inhibitor of leukemia cell proliferation, inducing their differentiation into macrophages, promotes TAM formation in the tumor microenvironment of various cancers. LIF levels correlate with increased TAM infiltration in GBM, prostate adenocarcinoma, thyroid cancer, and ovarian cancer [27]. Additionally, IL-10, a cytokine that induces a dominant immunosuppressive macrophage phenotype, was elevated in a 20% O_2_ environment with 5-ALA administration, but levels tended to decrease following irradiation and RDT, particularly in U251MG cells.

Nrf2, a factor activated by oxidative stress (e.g., ROS), plays a role in suppressing of inflammation. A study on HGG has shown that Nrf2 knockdown enhances the induction of ROS generation and cell apoptosis after irradiation [28]. In our study, Nrf2 levels were reduced by radiotherapy or RDT in both 20% and 3% O_2_ environments compared to the controls, supporting the idea that RDT induces ROS generation and apoptosis in both normoxic and hypoxic environments, contributing to anti-tumor effects.

CRT is a factor that promotes the uptake of tumor cells by macrophages, increases phagocytosis, and activates tumor immunity, whereas HMGB1 acts as an immunosuppressive protein functioning outside the cell. In this study, CRT and HMGB1 were expressed at higher levels in control and 5-ALA groups but tended to be suppressed by radiation exposure in both normoxic and hypoxic environments. HMGB1 has complex roles: it enhances radioresistance [29], promotes metastasis and proliferation, yet also exhibits tumor suppressive effects [30]. HMGB1 is released extracellularly from tumor cells following apoptosis and necrosis induced by radiation therapy, chemotherapy, and other treatments. To better understand the HMGB1 results obtained by ELISA in this study, further investigation into its correlation with the tumor immune environment using in vivo experiments is needed.

### 4.3. RDT Potential Under Hypoxia

Pathological findings indicate that brain tumors exist in a persistently hypoxic environment due to angiogenesis failing to keep pace with tumor growth [13]. It has been suggested that hypoxia in brain tumors promotes active migratory capacity, increases invasiveness, and confers resistance to both chemotherapy and radiotherapy [13,14,15]. The results of our study indicate that the anti-angiogenic effects of RDT on the tumor microenvironment under hypoxia may help to reduce treatment resistance.

HIF-1α and VEGF, markers of angiogenesis, are known to be highly expressed in hypoxic environments. HIF-1α, stabilized under hypoxia, is a transcription factor regulating the expression of approximately 60 genes involved in several cellular pathways, including glycolysis, angiogenesis, invasion, and epithelial–mesenchymal transition [31]. It influences VEGF and VEGF receptor expression, promoting abnormal blood vessel formation and further exacerbating the hypoxic cycle [32,33]. Additionally, HIF-1α has been found to affect PD-L1 expression [34], and reduced HIF-1/VEGF expression has been associated with improved prognosis in GBM [35].

In this study, HIF-1α and VEGF genes and their proteins were elevated in a 3% O_2_ environment, yet RDT suppressed the increase in soluble HIF-1α and VEGF proteins. A recent study demonstrated that soluble HIF-1α is measurable in rat plasma and that an increase in local VEGF gene expression in the carotid artery is consistent with HIF-1α plasma levels. [36]. We hypothesize that hypoxic stimulation promotes HIF-1α gene expression but does not promote the expression of downstream genes, such as the VEGFA gene, because RDT reduces the function of the stable HIF-1α protein as a transcription factor in the nucleus. VEGF suppression may reduce treatment resistance in HGG in a hypoxic environment and may contribute to improved prognosis.

The negative impact of hypoxia on RDT is that it may not improve the reduced immune response or the enhanced EMT state. RDT combines radiation and photosensitive substances to generate ROS, directly destroying tumor cells. However, in a hypoxic environment, ROS production is limited, and hypoxia promotes the proliferation of tumor-associated macrophages (TAMs) and regulatory T cells (Tregs). Additionally, the levels of immunosuppressive cytokines, such as IL-10 and TGF-β, increase, suppressing immune cell activation [36]. Even if tumor cells are destroyed by radiation-related immune responses (or radiation immunotherapy), the activation of the immune system may remain insufficient, resulting in limited long-term therapeutic effects. Consequently, radiation immunotherapy alone may not be able to overcome the reduced immune response. Furthermore, hypoxia promotes HIF-1α activation, which upregulates EMT-related genes, such as Snail and Twist, further enhancing EMT [37,38]. Therefore, the effectiveness of RDT may be improved by eliminating hypoxia or by combining it with treatments targeting EMT.

To eliminate the hypoxic state and enhance RDT efficacy, several combination treatments may be considered (Figure 6). First, reducing tumor volume through surgery is the most direct and reliable approach, as larger tumor masses inherently create hypoxic conditions. Anti-VEGF therapy before RDT is another promising option. Tamura et al. reported that all six GBMs treated with bevacizumab (BEV) exhibited reduced microvessel density compared to 11 BEV-untreated (control) GBMs. Furthermore, HIF-1α or CA9 expression decreased in five of the six treated tumors, whereas reduced expression of these markers was observed in only one of the 11 control GBMs [38].

Another strategy involves directly supplying oxygen to the tumor to oxidize the hypoxic environment. For example, studies have shown improved PDT efficiency using perfluorocarbon (PFC) nano-carriers, which enhance oxygen delivery [39], suggesting similar benefits for RDT. To inhibit EMT, the use of molecularly targeted drugs aimed at key EMT transcription factors may reduce cancer cell motility and treatment resistance [40,41]. Additionally, combining RDT with PD-1/PD-L1 inhibitors or CTLA-4 inhibitors could help reduce the immunosuppressive environment and simultaneously suppress EMT [42,43]. By facilitating immune recognition of tumor cells, such combination treatments may enhance the therapeutic effects of RDT.

To maximize the effectiveness of RDT, it is crucial to alleviate the hypoxic state of the tumor microenvironment and inhibit EMT progression. Combining RDT with oxygen delivery systems and molecularly targeted therapies may address these challenges and significantly enhance the therapeutic impact of RDT.

This study has some limitations. This study was conducted using ordinary cell lines, so we need to conduct a study to confirm the reproducibility of the results using glioma cell lines established from clinical samples in the future study. The experimental system lacks functional immune components. So, interpretation of cytokine dynamics remains limited. While RNA-seq analysis identified key regulators (e.g., DDIT4, VEGF, EMT markers), protein-level validation was lacking or insufficient for some targets. To confirm these issues, additional experiments are needed in the future work. Detailed in vivo studies are also needed to confirm the results obtained in this study. Another limitation is that this study did not consider p53 status of glioma cells. p53 mutations have been suggested to potentially affect the efficacy of PDT [44] and immunotherapy [16]. Future research should investigate the relationship between p53 status and RDT efficacy.

## 5. Conclusions

RDT appears to induce changes in the expression of various molecules, including cytokines and immune-related factors, thereby affecting the tumor microenvironment, particularly the micro-immune environment and angiogenesis-related factors. These effects differ depending on oxygen concentration. For instance, RDT enhances immune responses, potentially creating a synergistic effect with tumor-killing immune cells. Although high PD-L1 expression due to 5-ALA administration or RDT under normoxic conditions may contribute to immunotherapy resistance in vivo, the observed suppression of sPD-L1 elevation with RDT suggests potential synergy when combined with immune checkpoint inhibitors.

Conversely, hypoxia diminishes tumor-killing effects in proliferation assays in vitro due to reduced ROS levels and increases EMT-related molecules, rendering the “hypoxic environment” itself a treatment-resistant factor. Effective RDT may thus require addressing hypoxia through methods such as intensive tumor removal or anti-VEGF therapy. Promisingly, RDT itself may reduce VEGF expression even under hypoxia, helping to suppress neovascularization and tumor growth.

By continuing to establish foundational evidence of the usefulness of RDT, elucidating tumor resistance mechanisms, and developing combinatory treatments to overcome them, we anticipate further improvements in patient life expectancy and progression-free survival.

## Figures and Tables

**Figure 1 cancers-17-03927-f001:**
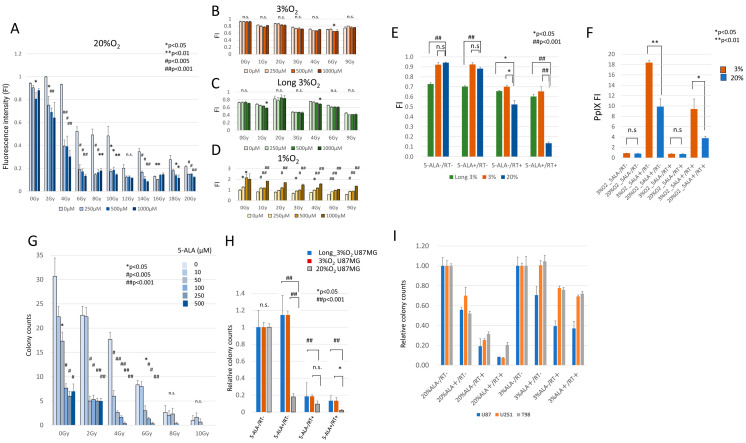
RDT effects depend on oxygen concentration, 5-ALA concentration, and X-ray dose. (**A**). RDT effects in a 20% O_2_ environment (normoxia). U87MG cells cultured under normoxia (U87MG_normox_ cells) were exposed to 0, 250, 500, or 1000 μM of 5-ALA, followed by X-ray irradiation at 2–20 Gy (1–10 exposures of 2 Gy). Cell numbers were quantified by measuring absorbance with the Cell Counting Kit-8. (**B**). U87MG cells initially cultured in a 20% O_2_ environment were transferred to a 3% O_2_ environment (U87MG_3%hypox_ cells) following 5-ALA administration and then irradiated with X-rays at doses of 1–9 Gy (single exposure for 1–3 Gy, two exposures for 4 Gy, and three exposures for 6–9 Gy). (**B**–**D**). The same experiments were conducted using U87MG cells cultured long-term in a 3% O_2_ environment over 6 months (U87MG_long_hypo_ cells) and short-term in a 1% O_2_ environment (U87MG_1%hypox_ cells). (**E**). Repeated experiments were performed using U87MG_normox_, U87MG_3%hypox_ and U87MG_long_hypo_ cells, treated with or without 1000 μM of 5-ALA and/or 6 Gy of X-rays (3 exposures of 2 Gy each). (**F**). Intracellular PpIX induced by 5-ALA and RDT was investigated in U87MG_normox_ and U87MG_3%hypox_ cell culture medium was replaced with DMEM containing 0 or 1000 μM 5-ALA, and cells were incubated in either 20% O_2_ or 3% O_2_. Four hours after 5-ALA addition, cells were either irradiated with X-rays at 2 Gy once a day for three days (6 Gy in total) or left unirradiated. Then, the fluorescence intensity at 625 nm in the cell lysate was measured using a microplate reader with an excitation wavelength of 415 nm. (**G**). Colony formation in U87MG_normox_ cells across different doses of X-ray irradiation (0–10 Gy) and varying 5-ALA concentrations (0, 50, 100, 200, and 500 μM). The number of colonies formed was counted after 14 days of incubation following the end of irradiation. Colony counts decreased significantly with increasing X-ray doses, particularly in the presence of higher 5-ALA concentrations, indicating a concentration-dependent enhancement of RDT efficacy in normoxia. At higher X-ray doses (8–10 Gy), the irradiation effect was the strongest and masked the presence of 5-ALA, making the differences between the numbers of colonies insignificant. Significant differences in colony reduction in 0 to 6 Gy groups are marked (* *p* < 0.05, ** *p* < 0.01, # *p* < 0.001). (**H**). Relative U87MG_normox_, U87MG_3%hypox_ and U87MG_long_hypo_ cells colony counts. The cells were divided into four treatment groups: untreated unirradiated control (5-ALA−/RT−), 1000 μM 5-ALA administration without irradiation (5-ALA+/RT−), 6 Gy irradiation only (5-ALA−/RT+), and combined treatment with 5-ALA and X-ray irradiation (5-ALA+/RT+, or RDT). Radiation was administered at 2 Gy once daily for three days. In the U87MG_normox_ cells, the combined treatment (5-ALA+/RT+) led to a substantial reduction in colony formation compared to 5-ALA, irradiation only, or untreated unirradiated control. In both U87MG_3%hypox_ and U87MG_long_hypo_ cells, the influence of both irradiation and RDT was reduced, with the less prominent effect in the U87MG_long_hypo_ cells, and the difference between 5-ALA−/RT+ and 5-ALA+/RT+ was less obvious compared to that in normoxia, suggesting limited effectiveness of irradiation and RDT under hypoxic conditions. (**I**). Similar results were observed in U251MG cells and T98G cells with or without 1000 μM 5-ALA administration and/or 6 Gy irradiation.

**Figure 2 cancers-17-03927-f002:**
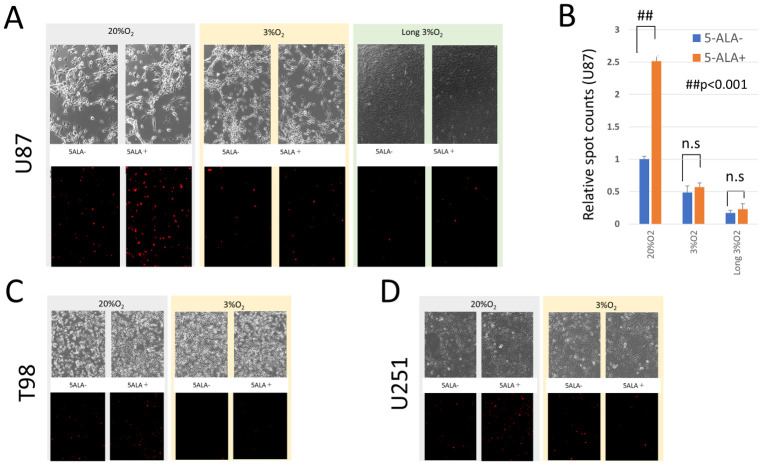
Fluorescence staining of ROS in normoxic and hypoxic environments after 5-ALA administration. (**A**). U87MG_normox_, U87MG_3%hypox_ and U87MG_long_hypo_ cells were incubated for 4 h in a culture medium either without 5-ALA (5-ALA−) or with 1000μM 5-ALA (5-ALA+). The upper panels show phase-contrast images, while the lower panels show red fluorescence, indicating ROS (singlet oxygen) levels in the cells. In the U87MG_normox_ cells (left column), 5-ALA administration led to a noticeable increase in singlet oxygen (visualized in red fluorescence) compared to the control without 5-ALA. Conversely, in U87MG_3%hypox_ and U87MG_long_hypo_ cells (middle and right columns), the increase in ROS was minimal, even with 5-ALA administration. (**B**). Relative spot counts of the singlet oxygen obtained from 3 areas. (**C**,**D**). In T98G and U251MG cells, as in U87MG cells, an increase in ROS was observed following 5-ALA administration under normoxic conditions compared to hypoxic conditions.

**Figure 3 cancers-17-03927-f003:**
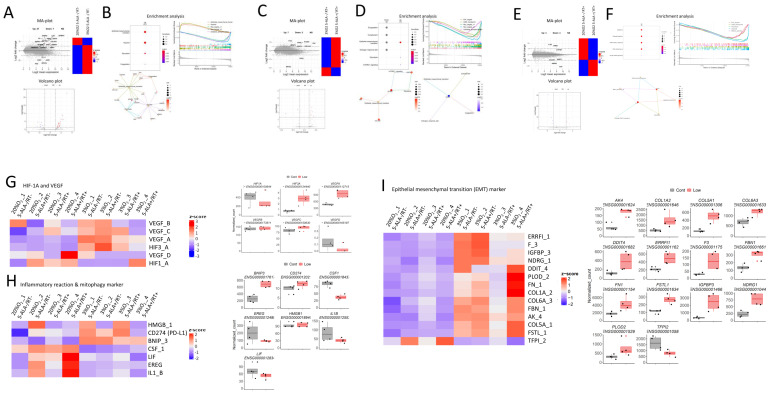
Results of RNA-seq differential expression analysis and RNA sequence heatmaps of ischemia- and angiogenesis-related genes, inflammatory and mitophagy-related genes, and EMT-related genes MA-plot, volcano plot, and enrichment analysis comparing different treatment groups of U87 cells under normoxic (20% O_2_) and hypoxic (3% O_2_) conditions. (**A**,**B**): Comparison of untreated U87 cells (5-ALA−/RT−) cultured in normoxia and hypoxia. (**C**,**D**): U87 cells treated with irradiation alone (5-ALA+/RT−) or with RDT (5-ALA+/RT+) under hypoxic conditions. (**E**,**F**): U87 cells treated with irradiation alone or with RDT under normoxic conditions. RNA-seq data were analyzed using RNAseqChef, a web-based application for automated, systematic, and integrated RNA-seq differential expression analysis (https://imeg-ku.shinyapps.io/RNAseqChef/, accessed on 15 May 2025). This tool, developed by Etoh K. and Nakao M., highlights gene expression features and provides insights into cell/tissue type-dependent actions ( https://doi.org/10.1016/j.jbc.2023.104810, accessed on 15 May 2025)). Heatmaps displaying RNA-seq data for representative sets of genes related to ischemia and angiogenesis (**G**), inflammatory response and mitophagy (**H**), and epithelial–mesenchymal transition (EMT) (**I**). Left: RNA was extracted from U87MG cells that were untreated, treated with 5-ALA (1000 μM), irradiated (6 Gy), or received combination treatment. Cells were cultured for 7 days in a 20% O_2_ environment (Cont1–4 group) and a 3% O_2_ environment (Low1–4 group). Gene expression levels were normalized using the transcripts per million (TPM) method and displayed as color-mapped heatmaps. Right: Box-and-whisker plots compare the median gene expression values between the normoxic group (Cont1–4) and the 3% O_2_ group (Low1–4), with statistical significance assessed using the Mann–Whitney U test.

**Figure 4 cancers-17-03927-f004:**
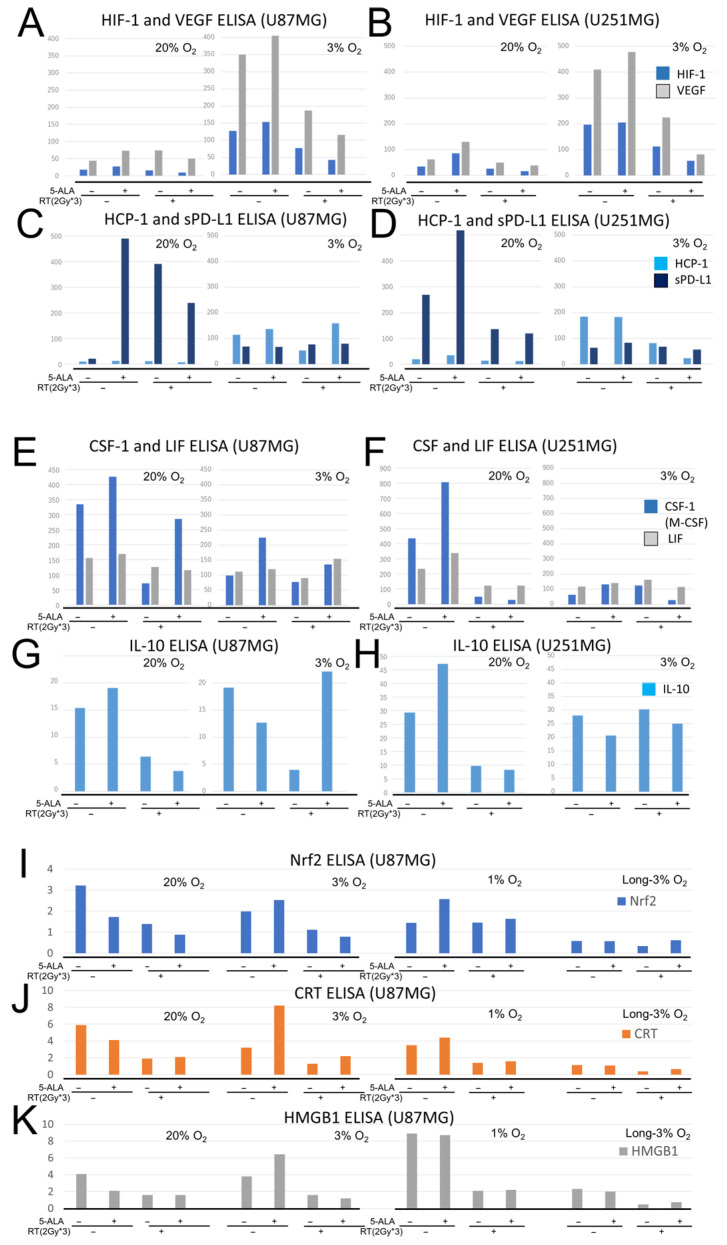
ELISA analysis of ischemia- and angiogenesis-related proteins, inflammatory and mitophagy-related proteins, anti-inflammatory protein, and tumor immunity activators after RDT. (**A**–**D**). Concentrations of HIF-1α, VEGF, HCP-1, and sPD-L1 in cultured U87MG cells (**A**,**C**) and U251MG cells (**B**,**D**) treated with 5-ALA and/or 6 Gy irradiation under 3% or 20% O_2_ conditions. Protein concentrations are given in ng/mL for HIF-1α, HCP-1, and PD-L1, and in pg/mL for VEGF. (**E**–**H**). Concentrations of CSF-1, LIF, and IL-10 in the culture supernatant of treated cells. The 5-ALA group received 1000 μM of 5-ALA, and the X-ray group was irradiated with 2 Gy three times (6 Gy in total). After irradiation, the supernatant was collected, and each protein concentration was measured using ELISA. Protein concentrations are given in pg/mL. (**I**–**K**). Ratios of Nrf2, CRT, and HMGB1 concentrations in the supernatant of four treatment groups of U87MG cells under various culture conditions with or without 5-ALA and X-ray irradiation. The doses of 5-ALA and X-ray irradiation in the groups were the same as described above. The ratio of protein concentration to cell number was calculated, setting the ratio of the negative control (no 5-ALA, no RT) at 20% O_2_ to 1.

**Figure 5 cancers-17-03927-f005:**
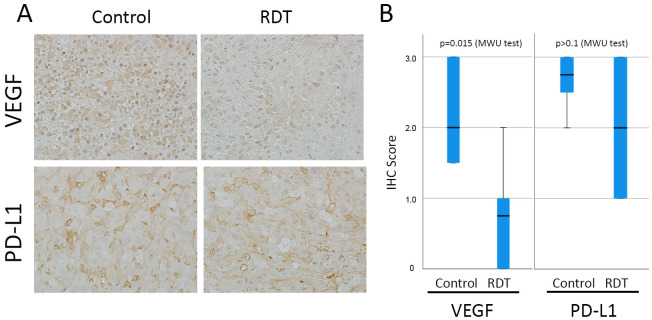
(**A**). Photographs showing immunohistochemical staining for VEGF and PD-L1 after in vivo experiments with or without RDT on U87MG tissues transplanted into mouse brains. Groups were divided based on the presence or absence of 5-ALA and X-ray RDT. The left side shows the control group without 5-ALA or X-ray, while the right side shows the RDT group with 5-ALA and X-ray. The upper panel shows VEGF staining, and the lower panel shows PD-L1 staining. In the RDT group, PD-L1 expression was not suppressed, whereas VEGF staining was suppressed. (**B**). Six randomly selected fields from the immunohistochemically stained tumor specimens were selected, and staining intensity was scored. RDT significantly suppressed VEGF expression.

**Figure 6 cancers-17-03927-f006:**
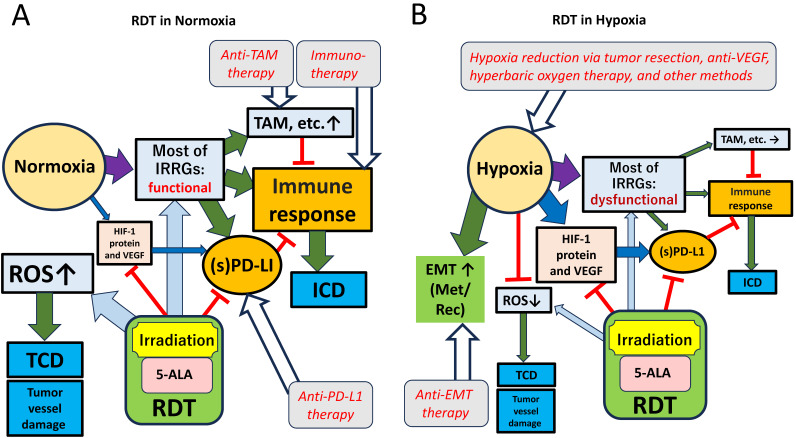
RDT for high-grade gliomas under in normoxic and hypoxic environments. (**A**). Effects on gene and protein expressions in tomor microenvironment when RDT is performed under normoxia. (**B**). Effects on these expression under hypoxia. The standard arrow indicates a positive effect, while the ⊢ arrow indicates a negative effect. EMT: Epithelial–mesenchymal transition, ICD: immunogenic cell death, IRRGs: inflammatory response-related genes, Met/Rec: metastasis/recurrence, RDT: radiodynamic therapy, ROS: reactive oxygen species, TAM: tumor-associated macrophages, TCD: treatment-induced tumor cell death.

## Data Availability

Raw data were generated at the Department of Neurosurgery, University of Tsukuba. Derived data supporting the results of this study are available from the corresponding author (E.I.) on request.

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
