# Peer review of "Radiodynamic Therapy for High-Grade Glioma in Normoxic and Hypoxic Environments for High-Grade Glioma"

_cancers, 2025, doi:10.3390/cancers17243927_

Round 1

Reviewer 1 Report (Previous Reviewer 3)

Comments and Suggestions for Authors

The authors have made substantive revisions in response to the prior review. Further, they acknowledge the limitations of their work and appreciate the future directions in preclinical studies. No further revisions are requested.

Reviewer 2 Report (Previous Reviewer 1)

Comments and Suggestions for Authors

The manuscript is fine. No further comments

This manuscript is a resubmission of an earlier submission. The following is a list of the peer review reports and author responses from that submission.

Round 1

Reviewer 1 Report

Comments and Suggestions for Authors Since the chages requested by the reviewers were considered in the manuscript, you should go ahead and accept the mansucript.

Reviewer 2 Report

Comments and Suggestions for Authors In my view, the authors did not adequately respond to the reviewer’s criticism. Since the intial votes were twice major revision and once reject and the authors did not add further experiments I would reject the manuscript.

Reviewer 3 Report

Comments and Suggestions for Authors

Yamada et al present a study regarding radiodynamic therapy (RDT) for adult high-grade glioma, assessing outcomes in vitro under normoxic and hypoxic conditions.

Major concerns:

  1. Although the manuscript is well-written and has partly acknowledged the concerns of the three prior external peer reviews by improving the section of the revised Discussion about study limitations and future directions, the study remains too preliminary and incomplete.
  2. The vast majority of the experiments shown in Figures 1-4 were done in one HGG cel line, the U87MG cell line. This cell line model of HGG is not very representative of adult glioblastoma in many ways, especially because it is p53 wild-type. Furthermore, U87MG is non-invasive in vivo when transplanted orthotopically (intracranially) in immunocompromised mice. The addition of U251MG, which is p53 mutant, only assisted to validate findings via ELISA experiments (Figure 5). Hence, the study’s conclusions, for all intents and purposes, are based on a cell line that the neuro-oncology community currently accepts as useful to initially test hypotheses and the conclusions require validation in vitro (and with all assays including bulk RNA sequencing) in other HGG cell lines, including patient-derived cell lines which are widely available, and in vivo using orthotopic transplantation (which the authors acknowledge in the revised Discussion).
  3. The removal of the non-orthotopic (subcutaneous HGG cell line implantation/transplantation) studies have actually diminished this work further. In fact, comparing the relatively normoxic environment of the non-orthotopic transplant to relatively hypoxic environment of the intracranial orthotopic environment could be quite informative.

Specific Concerns:

  1. 3% hypoxia adaptation likely represents chronic hypoxia rather than acute hypoxia. This requires further explanation in the Introduction or Discussion.
  2. The impact of mutant p53 versus wild-type to distinguish responses of HGG models and perhaps patients with HGG to RDT and PDT should be added to the Discussion.

Minor Concerns:

  1. Line 294: mTOR is “mammalian” target of Rapamycin, not “mechanistic”.
  2. Line 517: Change to Figure 6, since Figures 6 and 7 of the prior version were removed at the time of the initial manuscript revision.